# Downside of Helping Professions: A Comparative Study of Health Indicators and Health Behaviour among Nurses and Early Childhood Educators

**DOI:** 10.3390/healthcare12080863

**Published:** 2024-04-20

**Authors:** Melinda Csima, Judit Podráczky, Szabolcs Cseh, Dávid Sipos, Sára Garai, Judit Fináncz

**Affiliations:** 1Institute of Education, Hungarian University of Agriculture and Life Sciences, H-7400 Kaposvár, Hungary; podraczky.judit@uni-mate.hu (J.P.); financz.judit@uni-mate.hu (J.F.); 2MTA-MATE Early Childhood Research Group, H-7400 Kaposvár, Hungary; 3Doctoral School of Health Sciences, Faculty of Health Sciences, University of Pécs, H-7621 Pécs, Hungary; cseh.szabolcs@live.com (S.C.); sgarai.bh@gmail.com (S.G.); 4Department of Medical Imaging, Faculty of Health Sciences, University of Pécs, H-7621 Pécs, Hungary; cpt.david.sipos@gmail.com

**Keywords:** health condition, health behaviour, mental health, professionals working in early childhood education, healthcare professionals, nurses

## Abstract

The activities of health care workers and early childhood educators have received increased attention both in lay public discourse and in scientific discourse. These professional groups play a significant role in shaping the health behaviours of those they interact with; thus, understanding the patterns they convey is of paramount importance. The aim of our study is a comparative analysis of health conditions and health behaviours of professionals working in Hungarian early childhood education and nurses working in the healthcare system (n = 1591). We carried out our quantitative, cross-sectional research using convenience sampling among healthcare professionals working in nursing job positions (n = 581) and as early childhood educators (n = 1010), in south-west Hungary. Diagnosed chronic illnesses affect early childhood educators at a significantly higher rate (*p* < 0.05): the prevalence of musculoskeletal disorders is particularly high among them, as a result of which they reported a significant degree of physical limitation in relation to work. In the context of mental health, comparing the professional groups, nurses’ indicators were significantly (*p* < 0.001) more unfavourable in all examined dimensions. Moreover, the comparison in terms of educational attainment directed attention to the worse indicators of non-graduates. In this context, early childhood educators are less affected by all three dimensions of burnout (*p* < 0.001). As for health behaviour, the smoking habits of nurses are more unfavourable (*p* < 0.05). Regarding screening tests, participation in cytological testing was significantly higher among nurses, whereas early childhood educators showed increased participation in mammography (*p* < 0.001). Our findings draw attention to the fact that early childhood educators are primarily affected by chronic musculoskeletal disorders, while healthcare workers are more affected by problems related to mental health. Mental well-being can be further endangered by the fact that both professional groups perceive low social appreciation for the work they carry out.

## 1. Introduction

As a result of socio-economic changes and advancements in medicine, there is an increasing demand for the work of professionals who provide human and social services to treat the physical, psychological, intellectual, and emotional problems of clients and to satisfy their needs.

During their daily work, helping professionals are in constant verbal and non-verbal interactions with their clients, which requires a strong emotional involvement on their part [1]. They primarily work with clients facing stressful situations [2], which may complicate their interactions and lead to unprocessed traumas. The constant alertness and emotional strain associated with their work increases the risk of burnout and both physical and mental fatigue [3,4,5,6]. In the long term, all of these threaten their health and well-being and also determine their professional well-being [3,7].

Among the fields of helping professions (medicine, nursing, education, social work, psychotherapy, psychological counselling, and coaching), this study focuses on nurses working in the health-care system and professionals working in early childhood education. The activities of educators and healthcare workers receive increased attention both in lay public discourse and in academic discussions. Because of their role model status, it is crucial to assess their health conditions and health behaviours, since both professional groups contribute significantly to shaping the health behaviour of the clients and children they interact with [8,9,10,11].

Both in the case of healthcare workers and educators, idealised expectations are formulated by clients and their relatives: lay people regard the activities as a vocation rather than an occupation, where no room for error is allowed. The daily emergence of new challenges, multitasking, a fast-paced work environment with time pressures, and a lack of control over tasks are constant stressors present in their lives [12,13]. This is further compounded by the fact that, in addition to the clients entrusted to them, the children’s parents and the patients’ relatives also demand attention. As a result of constant availability and alertness, the boundaries between work and personal life become blurred, which may lead to role conflict and emotional overload [14], exacerbated by the increasing labour shortage [15].

Despite the moral and legal responsibilities associated with their work, the decline of social, professional, and financial appreciation is increasingly noticeable [15,16,17], leading to a continuous erosion of the attractivity of teaching and healthcare careers. In addition to mental strain, a high level of physical demand is also present for the representatives of both professional groups: among early childhood educators, it is primarily related to the care of young children, while among nurses it is related to the care of patients and the satisfaction of their basic physiological needs. Furthermore, due to their workplace environments (kindergarten or day-care centres and hospitals), both professional groups are highly exposed to various infectious diseases [18].

Both professional groups play an indispensable role in the functioning of society. Additionally, they face similar challenges in their work, which affect their health status and health behaviour. These factors fundamentally determine the quality and effectiveness of their work. While numerous studies have separately examined the health status and health behaviour of each professional group [3,4,6,7,9,10,11,12,13,17,19], just one comparative study has been conducted so far [16]; however, this focused on burnout only, with a small sample size. Therefore, our research can be considered as filling a gap.

### 1.1. Specific Factors Affecting the Health Indicators and Health Behaviour of Early Childhood Educators

Previous studies aiming to explore the health indicators of educators were mainly conducted among school-teachers of older children [19,20,21,22]. Studies focusing on professionals working in early childhood education have spread in recent decades; however, these are mostly of a local nature and involve small sample sizes [3,4,23,24].

Early childhood educators take responsibility for groups of children in day-care centres and kindergartens; this entails considerable mental and physical challenges. Furthermore, children come from diverse socio-cultural backgrounds, so daily routines and habits in the family may differ significantly from those required by the institution. It is a complex task for educators to respect family values and work in accordance with the institutional expectations when it comes to working with families representing different values [25,26].

Working in large groups means continuous occupational noise exposure, which appears as a non-specific stressor and can even lead to functional changes in the cardiovascular system and a weakened immune system [27,28]. Working in a high-noise environment can lead to tinnitus and hearing loss, accompanied by reduced concentration, fatigue, and memory decline. Additionally, stress-induced headaches and other psychological changes can also occur. Furthermore, the elevated volume used by educators in a noisy environment may contribute to chronic upper-respiratory tract diseases [29]. In addition to occupational noise exposure, early childhood educators are also at risk of musculoskeletal disorders. Throughout their daily care activities, they constantly lift the young children entrusted to them, squat, kneel, and bend down, placing significant strain on the musculoskeletal system and potentially leading to the development of various pains and changes in the affected areas. In addition, they carry out a significant portion of their daily activities sitting on child-sized furniture [30]. As a result, musculoskeletal disorders are among the most common occupational health problems among educators working in kindergartens. The long-term mechanical strain causes significant problems, including neck and/or shoulder pain (NSP) and low back pain (LBP) [11]; furthermore, lower extremity musculoskeletal disorders (MSD) are caused by continuous and prolonged kneeling, bending, and squatting and have varying degrees of pain associated with them [31].

Relevant research reported lower levels of smoking and alcohol consumption among the educators surveyed, compared to the adult population of the given country [30,32].

Due to the role model nature of the profession, it is outstandingly important to examine the health behaviour of early childhood educators.

### 1.2. Specific Factors Affecting the Health Indicators and Health Behaviour of Nurses

Research on healthcare workers is partially focused on the entire health workforce, while the majority of studies aim to uncover the health status and health behaviour of specific professional groups within the field [33,34,35,36]. The main conclusion of research on the entire workforce population is that the health of professionals deteriorates continuously over the years they spend at work, which can also affect the quality of the work they perform [17]. There are several well-known reasons behind this phenomenon. Prolonged surgical interventions, sedentary work due to increasing administrative obligations, and the patient-moving duties of nurses put strain on the musculoskeletal system [9]; thus, similarly to those working in early childhood education, the incidence of NSP, LBP, and MSD are significant among nurses [37]. In addition, the multi-shift work schedule and on-call system can disrupt the natural biological rhythm, causing physical and psychological stress [38,39]. Besides this, inappropriate or pathological eating patterns may develop, resulting in a higher proportion of obesity. Moreover, the multi-shift work schedule and on-call care make it difficult to organise social relationships and family life [40]. 

It is also a well-known fact that the mental health indicators of healthcare workers are less favourable than those for social groups with similar sociodemographic characteristics [34,35]. Among healthcare workers, burnout and mental stress are present at a higher rate [13]; moreover, working in shifts can increase the risk of developing sleep problems and depression [41]. 

Studies on the health behaviour of healthcare workers highlight the fact that despite having adequate levels of knowledge due to their special expertise, they do not always make efforts to preserve or improve their own health. The proportion of smokers is higher among healthcare workers compared to the entire adult population, and this is particularly true for female nurses working in countries with more favourable economic indicators [42]. In terms of alcohol consumption, studies show that the prevalence of alcohol consumption is higher among healthcare workers compared to the entire adult population; however, there are significant differences between certain professional groups: doctors tend to consume alcohol more frequently, while among nurses the proportion of abstainers is higher [43,44].

### 1.3. Characteristics of Early Childhood Education and Nursing in Hungary

In Hungary, early childhood education primarily occurs in two types of institutions: Infants and toddlers receive day care (mainly in nurseries) from the age of 20 weeks to 3 years. After reaching the age of 3, it is mandatory for all children in Hungary to attend kindergarten, until they start school (at 6–7 years of age) [45]. 

In nursery care, qualified day care teachers deal with the daily care and education of young children (with two day-care teachers responsible for a group of up to 14 children, supplemented by an additional caregiver for every two groups). These teachers hold either secondary or post-secondary professional qualifications or a bachelor’s degree in early childhood education, while caregivers typically possess secondary vocational qualifications [8].

In kindergarten education, the work of kindergarten teachers is supported by caregivers and teaching assistants. Each kindergarten group can accommodate a maximum of 33 children (usually consisting of 20–30 children), with two kindergarten teachers (working in shifts, with some overlap) and a caregiver. Additionally, an extra teaching assistant is employed for every three groups. Kindergarten teachers hold a bachelor’s degree, while caregivers and pedagogical assistants possess secondary professional qualifications [46].

The qualifications of hospital nurses in Hungary exhibit high heterogeneity, similar to professionals working in early childhood education: nurses hold either secondary or post-secondary professional qualifications, or BSc or MSc degrees in the field of health sciences. Depending on their qualifications, they perform various tasks in hospitals: advanced practice nurses with MSc qualifications typically occupy departmental management positions, while nurses with BSc qualifications (in the field of nursing and patient care) play a significant role in the operational management of nursing processes alongside nursing tasks [47]. Nurses with secondary vocational qualifications spend much of their working time on basic nursing tasks.

## 2. Materials and Methods

The aim of our research was to conduct a comparative analysis of health conditions and health behaviours among healthcare professionals working in nursing job positions and professionals working in early childhood education (including kindergarten teachers, day-care teachers, teaching assistants, and caregivers) in Hungary. In our surveys, we were seeking answers to the following questions:What kind of differences can be observed in the health conditions of nurses and early childhood educators?What are the most common chronic illnesses among them?What differences are there in the health behaviours of the two professional groups? What characterises their willingness to participate in various screening tests? What is the extent of risk behaviour occurrence among them?What differences can be seen between the two groups regarding the incidence of burnout and depression?

During the development of the questionnaire, the following key issues came into focus related to the examined topic: questions concerning job position, education, and qualifications and questions about health condition and health behaviour (smoking, alcohol consumption, and screening tests). During the elaboration of the measuring tool, in addition to question sets used in international surveys on health conditions and health behaviours (see the 36-item Short Form Health Survey (SF-36), European Health Interview Survey (EHIS), Maslach Burnout Inventory (MBI), Beck Depression Inventory (BDI), and WHO Well-Being Index (WBI) for a more in-depth exploration of the topic), we incorporated our own questions in the questionnaire. 

The SF-36 is a scale that measures the general quality of life. Condensed into 36 questions, it assesses the opinion of the population aged 14 and above regarding their own health condition across 8 dimensions. The dimensions examined by the questionnaire are the following: (1) physical functioning, (2) role limitation due to physical health, (3) pain, (4) general health, (5) energy/fatigue, (6) social functioning, (7) role limitations due to emotional problems, and (8) emotional wellbeing. The questionnaire is widely used in research conducted both in health sciences and social sciences [48,49]. Scale values for each dimension can range from 0 to 100, where the closer the value is to 100, the better the quality of life or the quality factor measured by the respective dimension. 

EHIS is carried out every five years based on mandatory European Union legislation [50]. Through data collection, we acquire a comprehensive understanding of the health condition of the population aged 15 and above, the underlying factors influencing it, and temporal changes in health condition.

The 22-item MBI covers the dimensions of depersonalisation, emotional exhaustion, and personal accomplishment. For each item, the respondent can indicate on a seven-point Likert scale how characteristic the answer is for them. In the case of statements on personal accomplishment, a low score indicates a greater degree of burnout, while in the case of statements on emotional exhaustion and depersonalisation, a high score may indicate a greater degree of burnout [51,52,53].

BDI or its abbreviated version is a tool for measuring the severity of depression [54]. The 9-item abbreviated questionnaire enquires about typical symptoms of depression on a 4-point Likert scale, where a higher score indicates an increased occurrence of symptoms [55].

WBI is intended to examine the individual’s general well-being for the two-week period prior to completing the questionnaire. The scale values of the five-item measuring tool can range from 5 to 25 with a higher total score indicating a more favourable well-being value [56,57].

During the statistical analysis, we used absolute and relative frequency series and mean calculation, as well as mathematical statistical tests: correlation calculations, χ^2^ tests, and Mann–Whitney tests were applied to explore the relationships between variables. The internal reliability of the applied standardised questionnaires was checked using Cronbach’s alpha. The Cronbach’s alpha values for the examined variables ranged from 0.72 to 0.89. The data were analysed using SPSS 29.0 software, and *p* < 0.05 was considered significant.

### Characteristics of the Sample

The target group of the survey consisted of professionals working in early childhood education in Hungary (kindergarten teachers, day-care teachers, teaching assistants, and caregivers), as well as nurses working in two central hospitals in south-west Hungary. Their selection was performed with simple, non-random sampling. In the survey, respondents volunteered, and their anonymity was granted. The research was conducted in accordance with the principles of the Helsinki Declaration [58]. The research has an ethical license with the identifier ‘Scientific work 25/2019’ approved by the Regional Committee of Science and Research Ethics of the Hungarian Medical Research Council. The sample size was 1591 after recording and cleaning the data, of whom 1010 people worked in early childhood education and 581 people worked in hospitals. The largest group of professionals working in early childhood education consisted of day-care teachers (41%), 37% was the proportion of kindergarten teachers, 16% worked as caregivers, and 6% were teaching assistants. Due to the specificities of the profession, only female respondents were included in the subsample. The respondents’ highest levels of education were closely related to their job positions: while nearly 98% of those working as kindergarten teachers had a degree, the educational background of day-care teachers presented a higher diversity: the majority (55.9%) held post-secondary vocational qualifications; at the same time, some employees had bachelor degrees (20%) or secondary school qualifications (23.4%). Teaching assistants typically had secondary school qualifications (49.2%), whereas among caregivers, vocational school training (60.4%) was the most common. In the case of nurses working in the health-care system, the proportion of those with secondary school qualifications was the highest (47.3%); in addition, 29.5% of them performed their work with post-secondary vocational qualifications. 

The average age of the participants in the study was nearly 45 years (44.44 ± 10.239), which draws attention to the unfavourable age composition of workers in helping professions. Kindergarten teachers are particularly affected in this regard, with an average age of 48.4 years. Among respondents, teaching assistants are the youngest (average age: 38.1 years), while the average age of nurses working in health-care system was 43.04 years. On average, respondents had been working in their chosen field for 17.80 years (±12.38). The differences in the length of time spent in the profession were more differentiated for those working with young children: kindergarten teachers had been working with preschool-age children for an average of 24.3 years, whilst teaching assistants had only done so for 6.4 years. In the case of nurses working in the health-care system, we did not observe a significant difference.

## 3. Results

### 3.1. Subjective Indicators of Health Condition

Based on our findings, it can be concluded that two-thirds (66.6%) of early childhood educators and 56.8% of nurses rate their health as ‘good’ or ‘very good’ (χ = 24.36; df = 4; *p* < 0.001). The general health dimension of the SF-36 questionnaire, in addition to the assessment of self-rated health conditions, includes the subjective experiences and perceptions related to one’s own health. In the case of the latter, we did not find any significant differences in the examined professional groups, nor in the educational level (Table 1).

The dimension ‘physical functioning’ provides information on the limitations experienced when performing various physical activities (e.g., lifting, bending, and kneeling): in this regard, nurses’ indicators are more favourable, and there is no significant difference in terms of education level (Table 1). Fifty-nine percent of professionals working in early childhood education experience some limitations due to their health condition during strenuous physical activity (such as lifting), while in 31% of them the limitation occurs when bending forward, stooping, or kneeling. The indicators of nurses are more favourable for both variables: 32.1% of them during strenuous physical activity (χ^2^ = 111.15; df = 2; *p* < 0.001), while 20.5% of them feel limitations when bending forward, stooping, or kneeling (χ^2^ = 47.46; df = 2; *p* < 0.001) (Table 1).

The dimension ‘role limitation due to physical health’ provides information on the extent to which the respondents’ physical health affected their daily work and activities. Both the job position and the level of education show strong differentiation (Table 1). A significantly higher proportion of educators had to reduce time spent at work or other activities due to problems related to physical health in the four weeks prior to responding (*p* < 0.001). 

The ‘Pain’ dimension of the SF-36 questionnaire refers to the intensity of pain experienced during the four weeks prior to data collection and its work-related consequences, for which the nurses’ indicators were also more favourable. We believe that physical pain is a significant hindrance to work; therefore, we separately analysed some of the questions belonging to this dimension. First of all, we wanted to find out whether there were any significant differences between the particular professional groups for the analysed variables. Regarding the physical pain experienced in the last four weeks, a significant difference can be verified between the professional groups (χ^2^ = 66.82; df = 5; *p* < 0.001). While 31.4% of early childhood educators reported moderate or stronger physical pain, this was only 19.2% for nurses. In examining the impact of physical pain on work performance, we proceeded from the assumption that workers in both job positions experience similar degrees of limitation due to the high level of physical strain. Contrary to our expectations, there was a significant difference between the two professional groups (χ^2^ = 24.91; df = 4; *p* < 0.001): 25.4% of early childhood educators reported at least a moderate level of disruption, while it was merely 17% among nurses (Table 1). 

When examining the differentiating factors of the dimension ‘Role limitation due to emotional problems’, education level was found to be a significant influencing factor, in addition to the job position (Table 1). While indicators related to physical functions were more favourable for nurses, emotional factors proved to be more favourable for educators. It is particularly worrying that 62.8% of nurses said that in the four weeks prior to responding, due to their mental health problems, they did not perform their work or other activities as carefully as they used to. The same proportion for educators was 21.3% (*p* < 0.001).

### 3.2. Objective Indicators of Health Condition

The objective indicators of health condition primarily refer to diagnosed chronic illnesses, as well as the conditions and symptoms associated with them, such as overweightness, obesity, and reduced ability to work. In this regard, it can be stated that 60.6% (n = 585) of early childhood educators and 41.8% (n = 243) of nurses have a chronic disease or have been struggling with a long-standing health complaint, many of them with more than just one.

In the sample, the most common chronic illnesses were waist and back pain, high blood pressure, allergies, and other joint problems such as neck pain and arthritis. The diagnosed musculoskeletal disorders (MSD, NSP, and LBP) should be highlighted from the perspective that the examined group is considered to be at particular risk in this respect, compared to the Hungarian population as a whole. 

As for long-standing health complaints, it should be mentioned that 59.1% of the respondents struggled with physical pain during the four weeks prior to data collection. Among those reporting physical pain, a significantly higher proportion were early childhood educators (χ^2^ = 68.82; df = 5; *p* < 0.001): 31.4% reported pain of at least moderate severity, whereas the same proportion among nurses was 19.2%. In this context, more than half of the respondents (50.6%) indicated that physical pain limited their ability to perform their daily tasks, including work and housework tasks. Early childhood educators reported a significantly higher degree of limitation (χ^2^ = 24.91; df = 4; *p* < 0.001).

In relation to health condition, we calculated the respondents’ BMIs based on body weight and height, which provides information on the number of workers affected by overweightness and obesity. Among early childhood educators, body mass index (BMI) values showed a somewhat more favourable picture: the rate of overweight or obese persons is 42%, while this proportion is significantly higher for nurses, at 56.1% (χ = 54.59; df = 3; *p* < 0.001).

### 3.3. Indicators of Mental Health

To assess mental health, in addition to the WHO Well-Being Index and the shortened, 9-item Beck Depression Inventory, our measuring tool also incorporated the Maslach Inventory, which is suitable for measuring burnout. In the case of depersonalisation, emotional exhaustion, and the Beck Depression Inventory higher values indicate mental health problems; however, in the personal accomplishment dimension and on the WHO Well-Being Index, lower values indicate these problems. Comparing the professional groups, the indicators of nurses in all examined dimensions prove to be significantly (*p* < 0.001) less favourable (Table 2), and the comparison by educational level highlighted the more favourable indicators of graduates. Early childhood educators are less affected in all three dimensions of burnout. As for depression, it is noticeable that two-thirds of the sample show mild symptoms of depression regardless of job position; moreover, among nurses, the incidence of severe depression is also significant (6.7%).

The perceptions associated with a profession are largely determined by what the representatives of the profession think of the appreciation of their field [3,59]. Accordingly, we included two statements in our measuring tool, the first one related to the appreciation received from colleagues and the second related to social appreciation (Table 3). 

It can be established for both fields that the surveyed helping professionals perceive the social appreciation of their work to be low, despite the fact that their activities are essential for the health of the population and for the education and care of children. The results are more unfavourable for nurses, who have worse indicators for both items: they perceive a lower level of appreciation both from colleagues and society. In addition to job position, education level was also found to be a differentiating factor: graduates perceive a higher level of recognition both from their colleagues and from society. 

By arranging the examined mental health dimensions and the scale values assigned to the statements regarding the perceived appreciation related to the studied profession in a correlation table allowed us to identify significant correlations. (Table 4). The values measured on the WHO Well-Being Index exhibit a significant, negative relationship (*p* < 0.001) with the depersonalisation and emotional exhaustion dimensions of burnout, as well as with the degree of depression; a significant positive relationship (*p* < 0.001) was found between the personal accomplishment dimension of burnout and the perceived appreciation of colleagues and society. This means that a high level of burnout in the dimensions of depersonalisation and emotional exhaustion is associated with an increased occurrence of depression and a lower level of well-being. Besides this, it results in a lower sense of appreciation both from colleagues and society. With regard to personal accomplishment, we found that its high level correlates with a higher sense of appreciation, especially from colleagues. 

### 3.4. Indicators of Health Behaviour

Examining the smoking habits of early childhood educators, it can be observed that 21.4% of the subsample smoke, while the proportion of smokers among nurses is 50.1%, indicating a significant difference (χ = 163.76; df = 3; *p* < 0.001). Education level also proved to be a differentiating factor: the proportion of smokers is considerably higher (37.6% vs. 20.9%) among non-graduates (χ = 49.95; df = 3; *p* < 0.001). Regarding alcohol consumption, both for early childhood educators and nurses, occasional consumption was the most frequent response. Weekly drinkers are identified at about 1% in both professional groups. Overall, it can be concluded that regular alcohol consumption is not typical based on the responses of the studied groups, so no further analysis was carried out in this regard. 

Willingness to participate in screening tests provides important information on the person’s individual activity related to health maintenance and disease prevention [60]. In addition to the mandatory lung screening and occupational health laboratory tests, 56.4% (n = 822) of the responding women underwent cervical cancer screening within the past year, while 14.3% (n = 208) took part in this type of screening more than three years ago. In Hungary, participation in a mammographic screening test is free of charge every two years for women over the age of 45. During our study, regarding the affected age group, we measured a participation rate of 56.1%, which shows significant differences by job position (χ = 185.80; df = 4; *p* < 0.001): while 67.3% (n = 371) of women aged 45 and over working in early childhood education participated in a mammographic screening test in the two years prior to the survey, this rate among nurses was only 33% (n = 89). Furthermore, it should be noted that 61.3% (n = 165) of the women over 45 among the nurses included in the study had never participated in a mammographic screening test. As for willingness to participate by educational level, the indicators are more favourable for respondents with a degree: while 66.5% (n = 198) of graduates were over 45 years of age, only 50% (n = 260) of non-graduates took part in a mammographic screening test in the two years prior to data collection (χ = 32.69; df = 4; *p* < 0.001). Among the responding women, 56.4% (n = 822) participated in cytological testing in the year before data collection, and another 29.3% (n = 427) within three years. We found significant differences by job position: 89.3% (n = 426) of nurses underwent cytological testing within three years, while 73.8% (n = 352) did so within one year (χ = 94.41; df = 4; *p* < 0.001). The same proportions are 84% (n = 823) and 48% (n = 470) for early childhood educators; these latter data are significantly lower than those of nurses.

## 4. Discussion

In recent years, increased attention has been focused on the health conditions and health behaviours of helping professionals [7,8,9,11]. In the case of those employed as educators, previous studies have concentrated mainly on teachers working in schools [19,20,21,22], while in the healthcare sector focus has been typically on individual professional groups [4,6,13,34].

The aim of this study was to comparatively analyse the health indicators and health behaviour of two professional groups whose representatives, as role models, influence the health behaviour of younger generations, clients, and their relatives. The comparison of the two fields is particularly justified because the physical and mental strain associated with their work is similar. However, in terms of physical strain, there are differences within the studied professional groups: while the nature of work-related physical activity is similar for all groups of educators working with young children, nurses exhibit a considerable degree of heterogeneity in this regard. For nurses, the level of mental and physical strain depends to a large extent on the workers’ job position and on the nursing needs of the patients they care for.

### 4.1. Health Condition 

According to previous studies, self-rated health is an adequate indicator of health condition [61]; to measure it, we incorporated the relevant question groups of the SF-36 questionnaire into our measuring instrument. According to the OECD ‘Health at a Glance’ report [62], 60.6% of the Hungarian population over the age of 15 rated their health as ‘good’ or ‘very good’, while the same proportion was 68.1% on average in OECD countries.

In terms of physical functions, limitations, and the occurrence of pain, our study emphasises the less favourable indicators among early childhood educators. Our data on objective indicators of health condition (with particular reference to the number of chronic illnesses and the affected organ systems), in accordance with the results of previous research, confirm the physical complaints present during work in the field of early childhood education; these complaints increase with the number of years at work, especially those related to MSD [11,20,21,22]. MSD-related pains are also present among nurses according to our study, but the incidence is lower. While in the entire adult female population of Hungary this group of disorders caused health loss in 9% [63], our results show higher rates of exposure for both examined professional groups: 35.8% among early childhood educators and 29.1% among nurses. Health loss caused by musculoskeletal disorders is exacerbated by overweightness and obesity, one of the biggest physiological health risks, which affects 49% of the Hungarian female population [63,64]. In comparison, the body mass index (BMI) values show a more favourable picture among early childhood educators: the proportion of overweight and obese persons is lower than among nurses and the general population.

During the examination of health conditions, we also paid close attention to the exploration of mental health indicators. We believe this is particularly justified because, according to WHO data, mental illnesses, including depression, anxiety, and their consequences can be considered one of the most serious health problems today [65]. In terms of mental health, our findings on burnout, depression, and general well-being draw attention to the fact that nurses have a higher level of burnout and depression, and a lower sense of general well-being. Our findings regarding burnout agree with the results of Afuqaha & Alsharah’s (2018) study in Jordan, which also confirmed a higher level of burnout among nurses [16]. The more favourable burnout indicators of early childhood educators may be related to the emotional surplus resulting from working with children, whereas for nurses, the constant confrontation with human frailty, suffering, and mortality may mean increased mental strain in everyday life. The high prevalence of mild depressive symptoms, regardless of job position, may be related to the generally unfavourable depression indicators of the Hungarian population [66]; nevertheless, further research is necessary to explore the underlying reasons.

### 4.2. Health Behaviour

During the examination of health behaviour, in addition to certain elements of risk behaviour, we covered the willingness to participate in screening tests. 

One of the most common forms of health-damaging behaviour is smoking, which leads to the development of various diseases (cancer, cardiovascular, and respiratory diseases), and results in a significant disease burden from both individual and social perspectives. According to WHO estimates, approximately 8 million people worldwide die each year due to smoking, of which about 1.2 million deaths—including 65,000 children’s—can be attributed to second-hand smoking [67]. In Hungary, 25.8% of the adult population smoke regularly, which is significantly higher than the OECD average (18%). For women, these figures are slightly more favourable, but the 20% rate can still be considered high [62]. Several studies point out that, in addition to smoking, healthcare workers are characterised by a sedentary lifestyle and improper eating habits [42,68], which is confirmed by our study as well. Our results emphasise the particularly unfavourable indicators for nurses: the proportion of smokers among them is almost twice as high as among the entire Hungarian adult population. The latter figure is particularly worrying, as professionals working in the healthcare sector set an example for the population with their health behaviour, and on the other hand, due to their qualifications, they are fully aware of the harmful effects of smoking on health.

In Hungary, the mortality rate of malignant neoplasms is extremely unfavourable compared to European averages; besides this, it is the second most common cause of death. Its rate among women increased from 22% to 27% between 1990 and 2019 [69]. Many types of cancer can be detected in time through screening tests, including breast cancer and cervical cancer, which have a high incidence and mortality. In the case of breast cancer screening, a participation rate of over 70% can effectively reduce breast cancer mortality [70]. In 2014, the participation rate in mammographic screening in Hungary was 65% in a two-year coverage, which is considered low by European standards [71]; the underlying reasons may be the fear of tumorous diseases and defence behaviour [72]. In order to increase willingness to participate, it is necessary to disseminate knowledge more widely, to dispel negative attitudes towards screening tests, and to refute misconceptions. Regarding the examined professional groups, the participation rate in mammographic screening tests was only 56.1% among women over 45 years of age, with a significant contribution from the low participation rate of nurses (33%).

The participation rate of women aged 25–64 years in cytological testing among the Hungarian population was 52.6% in terms of three-year coverage and 24.3% for annual coverage in 2007 [73]. By 2014, the three-year coverage rate had increased to 60%, but this was still extremely low by European standards [71]. It should also be mentioned that the COVID-19 epidemic had a negative effect on the participation of Hungarian women in cytological testing: by 2021, the annual participation rate dropped to 17% [72]. More than half (56.4%) of the women involved in our study took part in cytological screening tests in the year before data collection, and an additional 29.3% took part within three years, which is significantly higher than the Hungarian average. 

Considering the secondary prevention objectives, further research is needed to answer the question of why the participation rate in cytological testing is significantly higher among nurses while participation in mammographic screening is significantly higher among early childhood educators. 

### 4.3. Limitations

Main limitations of the study are the applied method and the representativeness of the sample. The results of the questionnaire survey are based on self-report, so the effort to meet expectations and knowledge about health and health behaviour may influence responses. The statistical distribution of the sample by job position does not represent the relevant parameters of the population.

## 5. Conclusions

The working environment of professionals in both health-care and early childhood education contains physical, biological, and ergonomic hazards, as well as a significant psychosocial risk for employees. Early childhood educators and nurses are exposed to similar physical strains in their work; despite this, the health indicators of early childhood educators are significantly worse, especially regarding musculoskeletal problems, which lead to considerable health loss among them. In terms of mental health, it can be inferred that nurses are at a greater risk of burnout. The low level of burnout among early childhood educators may be due to the unique characteristics of their profession: continuous, positive emotional contact with young children could serve as a protective factor, a hypothesis that warrants further investigation. Despite nurses possessing a sufficient level of knowledge due to their education and profession, they do not consistently engage in practices to maintain and enhance their health. This is evident not only in low participation rates in mammographic screening tests but also in the high prevalence of smoking and obesity among them. 

Our findings underscore the importance of developing primary and secondary prevention initiatives tailored to the work-related risk factors for both professional groups.

## Figures and Tables

**Table 1 healthcare-12-00863-t001:** Average values and correlations of the examined dimensions of the SF-36 among the studied groups.

	General Health	Physical Functioning	Role Limitations Due to Physical Health	Pain	Role Limitations Due to Emotional Problems
Cronbach’s alpha value	0.72	0.85	0.77	0.73	0.89
Job position (mean ± SD)					
early childhood educators	63.93 ± 19.94	86.83 ± 15.62	83.39 ± 29.53	73.95 ± 26.22	78.68 ± 34.61
nurses working in health-care system	65.54 ± 15.87	92.46 ± 11.95	92.37 ± 19.44	79.25 ± 21.81	68.68 ± 24.95
z *	−0.49	−9.17	−6.14	−4.13	−11.12
*p* *	0.62	<0.001	<0.001	<0.001	<0.001
Education level (mean ± SD)					
non-graduates	64.33 ± 18.00	89.13 ± 15.19	88.24 ± 24.98	76.23 ± 24.21	72.65 ± 29.64
Graduates	65.01 ± 18.93	90 ± 12.52	85.59 ± 27.89	76.28 ± 25.37	77.32 ± 33.80
z *	−0.70	−1.14	−2.57	−0.14	−3.07
*p* *	0.47	0.25	0.01	0.76	0.002

Note. n = 1591. * Mann–Whitney U test.

**Table 2 healthcare-12-00863-t002:** The differences in mental health indicators according to job position and educational level.

	Depersonalisation	Emotional Exhaustion	Personal Accomplishment	WHO Well-Being	Beck Depression
Cronbach’s alpha value	0.79	0.83	0.81	0.72	0.82
Job position (mean ± SD)					
early childhood educators	1.62 ± 2.6	15.24 ± 9.44	40.36 ± 5.97	20.23 ± 4.23	11.70 ± 2.94
nurses working in health-care system	6.87 ± 6.04	24.29 ± 9.72	33.01 ± 8.61	18.99 ± 5.18	15.42 ± 5.67
z *	−21.09	−17.33	−17.47	−5.55	−12.80
*p* *	<0.001	<0.001	<0.001	<0.001	<0.001
Education level (mean ± SD)					
non-graduate	4.42 ± 5.42	20.32 ± 10.87	36.24 ± 8.60	19.58 ± 4.88	13.71 ± 4.79
graduate	3.07 ± 4.53	17.15 ± 9.65	38.75 ± 6.85	19.85 ± 4.39	12.67 ± 4.64
z *	−3.48	−3.75	−4.82	−0.30	−3.95
*p* *	0.001	<0.001	<0.001	0.76	<0.001

Note. n = 1591. * Mann–Whitney U test.

**Table 3 healthcare-12-00863-t003:** Differences in the perception of work-related appreciation according to job position and educational level.

	‘I Feel That My Workplace Activity is Highly Appreciated by My Colleagues’	‘I Feel That My Workplace Activity is Highly Appreciated by Our Society’
Job position (mean ± SD)		
early childhood educators	3.95 ± 0.84	3.04 ± 1.13
nurses working in health-care system	3.41 ± 1.14	2.57 ± 1.09
z *	−9.37	−7.76
*p* *	<0.001	<0.001
Education level (mean ± SD)		
non-graduates	3.65 ± 1.07	2.75 ± 1.15
graduates	3.87 ± 0.93	3.02 ± 1.10
z *	−2.80	−4.08
*p* *	0.005	<0.001

Note. n = 1591. * Mann–Whitney U test.

**Table 4 healthcare-12-00863-t004:** Correlation matrix of mental health indicators with correlation coefficients (r).

	Depersonalisation	Emotional Exhaustion	Personal Accomplishment	Beck Depression	Appreciation by Colleagues	Social Appreciation
WHO Well-Being	−0.17	−0.35	0.36	−0.36	0.20	0.20
Depersonalisation		0.59	−0.51	0.43	−0.29	−0.19
Emotional exhaustion			−0.53	0.49	−0.22	−0.24
Personal accomplishment				−0.32	0.30	0.19
Beck depression					−0.27	−0.22
Appreciation by colleagues						0.45

Note. n = 1591. Spearman correlation, *p* < 0.001.

## Data Availability

Data are contained within the article.

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
