# Peer review of "Downside of Helping Professions: A Comparative Study of Health Indicators and Health Behaviour among Nurses and Early Childhood Educators"

_healthcare, 2024, doi:10.3390/healthcare12080863_

Round 1
Reviewer 1 Report
Comments and Suggestions for Authors
Dear All,
It was with great interest that I read the paper devoted to a very important issue of health indicators and health behaviour among nurses and early childhood educators. Having analysed the article, I reached the following conclusions:
1) The content of the work is consistent with the aim of the work.
2) With regard to the Introduction section, the authors could point out the peculiarities of the two analysed groups that are specific only to Hungary as compared to other countries, if such peculiarities exist. In addition, a brief discussion of the organization of the functioning of both groups in the Hungarian context is suggested (e.g. when it comes to nurses, do they have to hold a university degree? Are there such professions as those of nursing assistant or advanced practice nurse?).
3) It might be advisable to indicate just one aim of the study.
4) The authors should provide information on whether they have obtained permission to use the SF – 36 questionnaire.
5) As far as the Results section is concerned, the authors should not refer there to the research literature. Such information should be moved to the Discussion section (e.g., lines 233, 238-240, 299 - 302).
6) The Discussion section lacks a clear reference to results obtained in other countries.
7) The Limitation section should be included in the Discussion section.
8) The authors might want to reconsider whether it is necessary to reference works published over 20 years ago.
Reviewer 2 Report
Comments and Suggestions for Authors
Dear authors,
your manuscript in nicely prepared and would be interesting for the general audience. However there is several things you have to address before it would be ready for publication.
Title is nice structured.
Structure of abstract lack of introduction, and you have to may presented findings decide for the most important.
Introduction - you presented a lot of different research work, but somehow you didn't stress what is scientific contribution of YOUR work. Which scientific gap you are aiming to fill.
Methodology - please state that you followed Helsinki declaration and reference it. Can you state Cronbach alpha for each instrument.
Results are nice presented, maybe sample description is not too detailed.
Discussion - please connect your Tables with other findings.
Conclusion nicely presented.
References are current and supportive to your research
Round 2
Reviewer 2 Report
Comments and Suggestions for Authors
No further comments.